# Standard Expected Years of Life Lost Due to Malignant Neoplasms in Poland, 2000–2014

**DOI:** 10.3390/ijerph16244898

**Published:** 2019-12-04

**Authors:** Małgorzata Pikala, Monika Burzyńska, Irena Maniecka-Bryła

**Affiliations:** Department of Epidemiology and Biostatistics, the Chair of Social and Preventive Medicine of the Medical University of Lodz, Żeligowskiego 7/9, 90-752 Lodz, Poland; monika.burzynska@umed.lodz.pl (M.B.); irena.maniecka-bryla@umed.lodz.pl (I.M.-B.)

**Keywords:** standard expected years of life lost, premature mortality, malignant tumours, trends, Poland

## Abstract

The aim of the study was an analysis of mortality trends due to malignant neoplasms in Poland. The study material was a database, consisting of 1,367,364 death certificates of inhabitants of Poland who died during the period 2000–2014 due to malignant cancer. To calculate years of life lost, the SEYLL_p_ index (Standard Expected Years of Life Lost per living person) was applied. We also calculated AAPC (Average Annual Percentage Change). The SEYLL_p_ index (per 10,000 population) due to malignant neoplasms in Poland in males decreased from 586.3 in 2000 to 575.5 in 2014, whereas in females it increased from 398.6 in 2000 to 418.3 in 2014. The greatest number of lost years of life in 2014 was attributed to lung cancer (174.7 per 10,000 males and 77.3 per 10,000 females), breast cancer in females (64.5) and colorectal cancer in males (39.0). The most negative trends were observed for lung cancer in females (AAPC = 3.5%) and for colorectal cancer (AAPC = 1.8%) and prostate cancer (AAPC = 1.6%) in males. Many lost years could have been prevented by including a greater number of Polish inhabitants in screening examinations, mostly targeted at malignant neoplasm, whose incidence is closely connected with modifiable risk factors.

## 1. Introduction

The early 1970s in Western Europe marked the beginning of the so-called “cardiovascular revolution”, which was a sequence of changes regarding health [1]. In Poland, similar changes started to be observed only after 1990 [2]. The process of increasing incidence of cardiovascular diseases, which had started at the beginning of the 1960s, was then stopped and mortality due to the above cause between 1991 and 2010 in Poland decreased from 499 to 271 per 100,000 population [3]. Those changes were one of causes of lifespan extension. In 1991–2010, the lifespan increased by 6.2 years in males and by 5.5 years in females. A consequence of the health progress, observed in the last two decades, was a change in the epidemiological model of morbidity and mortality in Poland [4,5]. Currently, similar to Western European countries, malignant neoplasms are becoming the most prominent reason for premature mortality, particularly in people below the age of 65 [6,7]. Female mortality due to cancers is twice as high as that caused by cardiovascular diseases. World Health Organisation (WHO) experts believe that one third of cancers could be avoided, one third could be successfully treated and, in one third of cases, the quality of life of cancer patients could be improved [8,9].

The modification of risk factors plays the greatest role in primary cancer prevention. Doll and Peto estimated in their study the proportion of deaths attributed to the following risk factors in the USA: tobacco (25–40%), alcohol (2–4%), diet (10–70%), sexual and reproductive behaviour (1–13%), occupational exposure (2–8%) and environmental pollution (1–5%) [10]. Another component of primary prevention is screening for early diagnosis of cervical cancer, breast cancer and colorectal cancer. Hepatitis B virus (HBV) vaccine and human papillomavirus (HPV) vaccine play an important role in cancer prevention as well [11].

More than 100 various oncological diseases are called malignant neoplasms. In males, lung, colorectal and prostate cancer constitute almost 50% of cancer-related mortality. With regards to females, breast, lung, stomach, cervical and ovarian cancer is responsible for 60% of deaths [12]. Lung cancer in males as well as lung and breast cancer in females greatly shape trends in cancer diseases in the Polish population [13]. Prognosticating a cancer risk is highly important for prophylactic and therapeutic purposes. An analysis of historical trends and prognosis, even short-term, allows evaluating the change in risk factors and the effectiveness of an intervention. This method also enables evaluating the effectiveness of screening studies. A comparison of the short-term cancer-related mortality prognosis with real rate values can be an invaluable asset in evaluation of screening programmes [14].

Currently, cancer is the second most common cause of death in Poland and Europe, responsible for 27.6% of deaths in Poland, 26.5% of deaths in European Union countries and almost 20% of deaths in the WHO European Region [15]. Over 10 years, the number of cancer-related deaths has increased by 6.3% and in 2011 it contributed to death of 1,281,000 inhabitants of 28 EU countries. Overall, 37.1% of deaths in the population below 65 years of age were attributed to cancer. The highest percentage was noted in Holland (48.0%), whereas the lowest in Finland (28.4%). Since the mid-1980s, the incidence of neoplasm has increased by 32% and mortality due to the above cause decreased by 10% [16,17].

The number of lost years of life in Poland due to premature mortality or disability, caused by neoplastic diseases, is one of the highest in Europe [18]. The standardised DALY measure for neoplastic diseases in Poland in 2013 was 1822 per 100,000. It was the highest in Hungary (2167). In France it was 1623, in Germany it was 1350, in Greece it was 1255 and was the lowest in Cyprus (794).

A direct result of premature mortality is the number of years of life lost by the Polish population. To evaluate health of a particular population, authors of foreign studies on epidemiology are more and more frequently using measures which express premature mortality in units of time lost. Many of them claim that the number of years of lost life is a more reliable measure of mortality than other, commonly used mortality rates, since it more effectively points out social and economic outcome of this phenomenon [19,20,21]. Measures of years of life lost enable evaluating progress in treating diseases, because they include information on a decrease in the number of deaths and extended survivability.

There are different approaches to methods of calculation of these measures, depending on the adopted life span limit. According to the PYLL (*Potential Years of Life Lost*) index, the life span limit ranges from 60 to 85 years. The life span limit is often controversial, which is a disadvantage of the index. Another disadvantage is ignoring benefits, arising out of health interventions targeted at the oldest social groups. Thus, the PYLL index, being less reliable, is not considered in analyses. A better measure is the so-called PEYLL (*Period Expected Years of Life Lost*) index, according to which a local period of expected life span in each age is considered the life span limit. It does not include arbitrarily adopted age, above which deaths are not considered for the purpose of calculating the burden. However, the PEYLL index cannot be used to make comparisons over time or to distinguish between societies, characterised with different expected life span, which is a negative aspect of the index. The SEYLL (*Standard Expected Years of Life Lost*) index does not have such a negative feature. Here, expected life span is adopted in order to calculate the number of years of life lost on the basis of an ideal standard [22].

The aim of the study was to evaluate mortality trends due to malignant neoplasm during the fifteen-year period, 2000–2014, with the application of standard expected years of life lost.

## 2. Materials and Methods

The research project was granted an approval of the Bioethics Committee of the Medical University of Lodz on 22 May 2012 No. RNN/422/12/KB.

The study material was a database including 5,601,568 death certificates of all inhabitants of Poland who died during the period 2000–2014. Of this number, 1,367,364 people died of malignant cancer. The data were provided by the Department of Information of the Polish Central Statistical Office. The procedure of coding causes of death in Poland is performed in a similar to the one carried out in the majority of countries in the world, based on the so-called primary cause of death, or the disease which triggered a pathological process, leading to death.

The authors calculated standardised death rates (SDR). The standardisation procedure was performed with the use of direct method, in compliance with the European Standard Population, updated in 2012 [23].

Years of life lost were counted and analysed by the method described by Murray and Lopez in Global Burden of Disease (GBD) [24]. The SEYLL index (*Standard Expected Years of Life Lost*) is used to calculate the number of years of life lost by the studied population in comparison to the years lost by the referential (standard) population.

There are some methods of calculating lost years of life and the main difference between them is a point of reference, i.e., the level of mortality which is considered “ideal”. In the Global Burden of Disease (GBD) study 2010, WHO experts recommend using life tables, based on the lowest noted death rate for each age group, in countries with population above 5 million [25].

In this study, the SEYLL index was calculated according to the following formula:$$SEYLL\;=\;\overset{l}{\underset{x\;=\;0}{\sum }}d_{x}e_{x}^{*}$$
where $e_{x}^{*}$ is life expectancy, based on GBD 2010 life tables, *d_x_* is number of deaths at age x, *x* is age at which the person died and l is last age which the population reaches.

The authors also calculated SEYLL_p_ (per living person) and SEYLL_d_ (per death) indices, where the SEYLL absolute number corresponded to the number of inhabitants and the number of people who died due to the analysed cause [26].

Death causes were coded according to the International Statistical Classification of Diseases and Health Related Problems—Tenth Revision—ICD-10. The study analysis included malignant tumours (C00–C97), such as: bladder cancer (C67), brain cancer (C70–C72), breast cancer (C50), cervical cancer (C53), colorectal cancer (C18–C21), leukaemia (C95), liver cancer (C22), lung cancer (C33,C34), ovarian cancer (C56), pancreatic cancer (C25), prostate cancer (C61) and stomach cancer (C16).

The analysis of time trends was carried out with *joinpoint* models and *Joinpoint Regression* program, a statistical software package developed by the U.S. National Cancer Institute for the Surveillance, Epidemiology and End Results Program [27]. This method is an advanced version of linear regression, where time trend is expressed with a broken line, which is a sequence of segments joined in joinpoints. At these points, the change of the value is statistically significant (*p* < 0.05). We also calculated *annual percentage change* (APC) for each segment of broken lines and *average annual percentage change* (AAPC) for a full range of analysed years with corresponding 95% *confidence intervals* (CI).

## 3. Results

Standardised death rates (SDR) due to malignant tumours during the period 2000–2014 in Poland decreased. In 2000, SDR was 47.24 per 10,000 males and 24.11 per 10,000 females. In 2014, SDR was 40.51 and 21.90 per 10,000 population, respectively (Table 1). In 2014, the highest SDR values observed in the male group regarded: lung cancer (11.72), colorectal cancer (5.20) and prostate cancer (4.11). As for the female group, they were the highest for: lung cancer (3.73), breast cancer (3.05) and colorectal cancer (2.56). With regards to two cancers, characterised with the highest mortality, i.e., colorectal cancer in males and lung cancer in females, SDR in 2014 was higher than SDR in 2000.

The number of standard expected years of life lost per 10,000 males (SEYLL_p_) due to malignant tumours in 2000 was 586.3 (Table 2). During 2000–2006, SEYLL_p_ values were growing and the annual percentage change (APC) was 0.4% during this period (Figure 1). In 2006, the index value started decreasing at a pace of −0.8% (*p* < 0.05), and in 2014 it was 575.5 per 10,000 males. During 2000–2014, the average annual percentage change (AAPC) was −0.3%.

In the female group, the SEYLL_p_ index in 2000 was 398.6 per 10,000 females. (Table 3). During the fifteen-year study period, the trend changed twice. During 2000–2007, SEYLL_p_ values were increasing at an annual pace of 0.7% (*p* < 0.05); during 2007–2011, they were decreasing at an annual pace of −0.7%; and from 2011, they were growing again, at an annual pace of 0.9%. AAPC for the period 2000–2014 was 0.3% (*p* < 0.05). In 2014, the SEYLL_p_ index was higher than in 2000, i.e., 418.3 per 10,000 females.

An analysis of particular cancers confirmed that lung cancer contributes to the greatest number of years of life lost in males. However, values of this index are steadily decreasing. In 2000, the SEYLL_p_ index was 196.1 per 10,000 males. During 2000–2007, the annual pace was −0.2%; since 2007, the annual decrease has been more rapid, i.e., 1.7% (*p* < 0.05). This decline trend in males also regarded the SEYLL_p_ index due to stomach cancer (APC = −1.4%, *p* < 0.05). With regards to leukaemia and liver cancer, SEYLL_p_ values were quite stable (APC = −0.3% and 0.2%, respectively). With regards to all remaining malignant tumours, the SEYLL_p_ index for the male group increased during the period 2000–2014. High values of the SEYLL_p_ index (29.5 in 2000, and 39.0 in 2014) as well as high AAPC values (1.8%, *p* < 0.05) were observed for colorectal cancer. On the other hand, APC dropped from 2.7% during 2000–2008 to 0.6% during 2008–2014 (Figure 2). Prostate cancer contributed to a gradual annual 1.6% increase. The SEYLL_p_ index per 10,000 males due to the above cause increased from 26.5 in 2000 to 34.5 in 2014. A stable increase was also observed for bladder cancer (AAPC = 1.3%, *p* < 0.05), pancreatic cancer (AAPC = 1.2%, *p* < 0.05) and brain cancer (AAPC = 0.6%, *p* < 0.05).

In the female group, the most negative trend was observed for lung cancer. The SEYLL_p_ index, which in 2000 was 46.5 per 10,000 females, was growing during the fifteen-year study period by 3.5% annually, and in 2014 its value was 77.3 (Table 3). The disease contributing the second highest number of years of life lost in females is breast cancer. In 2000, the SEYLL_p_ index was 57.9 per 10,000 females. During the period 2000–2007, it was increasing by 0.6% annually (*p* < 0.05), whereas, during 2007–2010, it was decreasing by −1.2% (*p* > 0.05). In 2010, the value started to increase again and the annual increase rate was 2.6% (*p* < 0.05). An increase trend was also observed for pancreatic cancer (AAPC = 1.6%, *p* < 0.05), colorectal cancer (AAPC = 0.9%, *p* < 0.05) and ovarian cancer (AAPC = 0.7, *p* < 0.05). With regards to the last-mentioned cancer group, since 2007, the SEYLL_p_ index has not been growing (Figure 2). Low SEYLL_p_ index values, but a rapid increase in the pace of these values, were observed for bladder cancer (AAPC = 2.7%, *p* < 0.05). A positive trend, i.e., a decrease in the number of years of life lost, was noted for cervical cancer (AAPC = −2.1%, *p* < 0.05), liver cancer (AAPC = −2.5%, *p* < 0.05), stomach cancer (AAPC = −1.6%, *p* < 0.05) and leukaemia (AAPC = −1.0, *p* < 0.05).

The authors revealed that the number of years of life lost by one person who died of malignant neoplasm (SEYLL_d_) was decreasing in both males and females. Each male who died due to a disease included in this disease group in 2000 lost on average 22.6 years of life. In 2014, this number was 20.3 years (AAPC = −0.8%, *p* < 0.05). With regards to females, the SEYLL_d_ index decreased from 21.5 in 2000 to 19.4 in 2014 (AAPC = −0.7%, *p* < 0.05). Values of the SEYLL_d_ index indicate that in 2014 the following diseases contributed to the highest number of lost years of life in males: brain cancer (27.7), leukaemia (21.4), pancreatic cancer (21.4), liver cancer (21.2), lung cancer (20.5) and stomach cancer (20.0). Each female who died in Poland in 2014 lost on average more than 20 years of life due to: brain cancer (24.7), cervical cancer (24.3), ovarian cancer (22.4), breast cancer (21.4) and lung cancer (20.9). A significant decrease in the SEYLL_d_ index was observed for almost all of the studied causes of death in both males and females (except for bladder cancer in females).

## 4. Discussion

According to the results of the GLOBOCAN study, the global cancer burden is annually about 18.1 million new cases and 9.6 million deaths [28]. In various parts of the world, people are affected by various kinds of cancer. Studies conducted by the American Cancer Society reveal that the most common kinds of cancer occurring in developed countries include prostate cancer, lung and bronchial cancer, colorectal cancer and breast cancer [29].

A study conducted in 60 countries across six continents during the period 2000–2010 confirmed a global decrease in neoplasm-related mortality. The average annual decrease was 1.2% for males and 0.8% for females. In most of the countries—41/60 for males and 35/60 for females—the annual decrease was 1.0% or above. A decrease was observed for the majority of the most common cancers, except for liver cancer, for which mortality indices were growing in half of the studied countries, particularly in Latin America and North America. Besides, in most countries, female mortality related to lung cancer increased [30,31]. These findings correspond to results obtained in previous studies. Data included in GLOBOCAN 2002 [32] and 2012 [33] indicate that the cancer-related mortality rate was decreasing in the study period at an annual pace of 1.0% in females and 0.8% in males. Consequently, the decreasing death rate resulted in a reduction of deaths by 814,000 during the period 2002–2012. During 1991–2010, in the United States, the annual average pace of change was −1.06% [34]. A decrease in APC values (APC ≥ −1.0%) was also noted in Europe [35] and other countries, including Asian, African and Latin American countries with an average and low income [36,37]. Apart from demographic characteristics, three other determining factors contribute to the occurrence of different death rates: (1) frequency of risk factors; (2) screening tests and other methods of early diagnostics; and (3) access to treatment. Mortality rates, almost exclusively associated with modifiable risk factors, mostly refer to lung, gastric and liver cancers. These mortality trends mostly depend on incidence rates and are less attributed by economic inequalities [38]. Similar male and female smoking patterns are translated into an increase in the lung cancer incidence in females, which entails an increase in the death rate due to this cause in the female subpopulation. Although the frequency of smoking tobacco decreased, the mortality rate due to lung cancer is becoming similar in males and females in some European countries, and we can expect that its value will be increasing in females all around the world [39].

Lung cancer is the most common cancer affecting men (it constitutes 20% of disorders and contributes to 30% of deaths). For almost two decades, the incidence of this disorder has been decreasing. This positive trend results from a decrease in tobacco smoking in males of all ages. A different trend was observed in females in whom active smoking is different due to cohort effect (date of birth in calendar time). The highest smoking level was observed in women born between 1940 and 1960; during certain periods, it was as high as 50%. In the population of females born after 1960, the frequency of smoking dropped by half and was 20–25%. Currently occurring malignant neoplasms affecting women of various ages are closely associated with their exposure to carcinogenic factors during a twenty-year latency period. The observed cohort effect indicates that this kind of cancer and cancer-related mortality are still on the increase and this growing trend will be observed for some time [40].

The study performed by the authors made the problem more evident when they revealed quite a considerable increase in the number of years of life lost due to lung neoplasms during 2000–2014. APC in this period was 3.5%. In Poland, tobacco attributable fraction is estimated to be around 80–90% in males and around 60–70% in females [41]. In non-smokers, the incidence of malignant lung neoplasm is very low, i.e., 5 per 100,000 population [42]. Experiments and epidemiological studies conducted in Poland and other countries confirm the effectiveness of eradication of tobacco in combating lung neoplasm. It is worth comparing time trends of mortality due to lung malignancy in young adult males in Poland and Hungary. In the 1960s and 1970s, the trends were almost identical. After initiating an anti-nicotine campaign in Poland in the 1990s, they became different. In Hungary, the incidence of lung malignancy was one of the highest in the world, whereas, in Poland, it started to decline. In Hungarian females, mortality rates were almost three-fold higher than in Poland [43].

Breast cancer is the most common kind of cancer in the group of women. It is the second most common cancer which affects the worldwide population. We observed decreasing mortality rates due to breast cancer, particularly in high-income countries of Europe. Favourable trends were recorded during 2002–2012 in EU countries. The standardised mortality rate decreased during this period from 17.9 to 15.2 per 100,000. A further decline, up to 13.4, is expected to occur by 2020 [44]. Even though Poland is a member of the EU and is one of the high-income countries, the results of this study show that the number of years of life lost due to breast cancer stopped growing in 2007, but, since 2010, a negative increasing trend has been observed again (APC = 2.6%). It is associated with an increasing number of deaths due to breast cancer. To explain these unfavourable trends in Poland, we should point out changes in the age structure of women. Data from the National Cancer Registry indicate that most breast cancer-related deaths occur after the age of 50 (90%) [45]. The percentage of women over 50 years old in Poland in 2010 was 37.8% and it rose to 39.3% in 2014 [46]. With regards to other countries, breast cancer-related mortality has started decreasing, or at least stopped increasing, in most countries all around the world. This positive trend is associated with advances in cancer treatment and implementation of breast cancer screening tests. However, a relationship between early diagnostics and screening tests and a decrease in breast cancer-related mortality is still disputable [47,48,49].

Malignant neoplasm of the prostate is the second most common neoplasm in Poland (almost 14,000 diagnosed cases and 4440 deaths in 2014). Wong et al. classified Poland together with Brazil, the Czech Republic, France, Ireland, Israel, Italy, Japan, The Netherlands, Spain, Switzerland and the UK into the group of countries with increasing incidence and decreasing mortality due to prostate cancer [50]. However, mortality in Poland in this group was decreasing the least rapidly. The most rapid decline was observed in Israel (AAPC = −3.9), France (AAPC = −3.9%) and the Czech Republic (AAPC = −3.8%). The most positive trends are noted in three countries, namely Finland, Sweden and the US, where both morbidity and mortality are diminishing. Prostate cancer mostly affects older age groups. The risk of death due to this cancer increases in the seventh decade of life [51]. Males over 70 years constituted 5.9% of the population in 2000, while, in 2014, the value was 7.6% of all men in Poland [46]. It can therefore be concluded that aging of the population of Poland is the main reason for the increase in prostate cancer mortality.

Poland belongs to the group of countries with increasing morbidity and mortality rates due to colorectal cancer. Similar trends can be observed in several European countries (Bulgaria, Russia, Belarus, Estonia, Lithuania, Croatia, Spain and Latvia). The most positive trends (decreasing morbidity and mortality rates) in Europe are noted in Austria, Czech Republic, Iceland and France [52]. The rises in incidence point to the influence of dietary patterns, obesity and lifestyle factors, whereas the mortality declines seen in more developed countries reflect improvements in survival through the adoption of best practices in cancer treatment and management in developed countries. Unfavourable incidence trends in Poland are probably resulted from the progressive adverse changes in the lifestyle and dietary patterns of Poles [53]. A continuous increase in the amount of alcohol consumed by Poles is also of concern: in 2000, it amounted to 7.1 L per capita, while, in 2014, it was 9.4 L [54]. Additionally, the reasons for a negative epidemiological situation related to colorectal neoplasms can be found in the ineffectiveness of the screening programme that has been carried out in Poland since 2000. Unfortunately, the response rate to those invitations is highly unsatisfactory. In 2014, only 17.4% of the invited respondents arrived for examinations [55].

Cervical cancer occurs relatively rarely in Poland. In 2012, around 2800 females developed the disease and 1700 females died of it. The incidence of cervical malignant neoplasms has been decreasing in Poland since the 1980s and in the last decade the decrease was even more rapid. Cervical cancer-related mortality has also been decreasing for a long time. It was noted by the authors of the study that this decline was also observed in the number of years of life lost due to the above disease. During 2000–2014, the SEYLL index was decreasing at an annual rate of −2.1%. However, the mortality level in Poland is still higher than in any Western European country. In addition, the rate of decline in mortality due to cervical malignant neoplasms is much more rapid in Denmark, Finland or Great Britain than in Poland, which is the reason for the increase in disparities [56]. The percentage of participants in cervical cancer screening examinations in Finland is 57%, in Holland is 70% and in UK is 85%, whereas in Poland in 2015 it was only 20% [57]. Obligatory vaccination against human papilloma virus (HPV) might to some extent decrease cervical cancer-related incidence and mortality. In countries where vaccination against HPV is commonly applied, the number of cases of cervical cancer decreased by half [58]. Unfortunately, currently, vaccines against HPV are only recommended in Poland but they are not financed by the Ministry of Health.

It is estimated that decreased stomach cancer-related mortality has mostly contributed to a decrease in the death rate due to neoplasms (APC = −2.7% in males and −2.8% in females). The most noticeable decline was observed in Asia, Latin America and post-Soviet countries. Decreased mortality was also observed in the US during 2004–2013 (APC = −1.1%). The authors of this study confirmed that the number of years of life lost due to stomach cancer decreased during the period 2000–2014. APC for males was −1.4%, and for females −2.5%. The decrease in the incidence of stomach cancer, observed for a few decades, is to a great extent associated with monitoring of *Helicobacter pylori* infections [59,60]. Fresh alimentary products have become widely available, people undergo screening tests and hygiene in countries much affected by stomach cancer has substantially improved [61].

This study has potential limitations. The quality of the analyses performed on the mortality statistics depend on the completeness and accuracy of the information contained in the death certificate and the proper and precise description of the cause of death. Poland is a country with 100% completeness of death registration. To standardise death causes, which are subject to further statistical analyses, it was determined that the doctor who pronounces death is responsible for filling out the death card, on which he or she puts the primary, secondary and direct death cause, whereas qualified teams of doctors are responsible for coding death causes according to the ICD-10 classification.

The data relating to 2012 show that the causes of more than 28% of deaths (about 109,000) were inaccurately described; however, in the majority of cases (78,500), it concerned deaths due to cardiovascular diseases [62]. Coding of malignant neoplasms does not raise any objections generally.

## 5. Conclusions

The analysis of changes in the number of years of life lost due to neoplastic diseases affecting inhabitants of Poland, conducted over 15 years, points out high values of the SEYLL_p_ index and negative trends regarding lung cancer in females, prostate cancer in males and colorectal cancer in both males and females. Since 2010, the number of years of life lost due to breast cancer in females has also been increasing. We could avoid such a great loss by including more inhabitants of Poland in screening tests, aiming at detecting those malignant neoplasms whose incidence is closely associated with modifiable risk factors. Besides, there is a need to implement programmes on health promotion, including smoking cessation, particularly targeted at young women.

## Figures and Tables

**Figure 1 ijerph-16-04898-f001:**
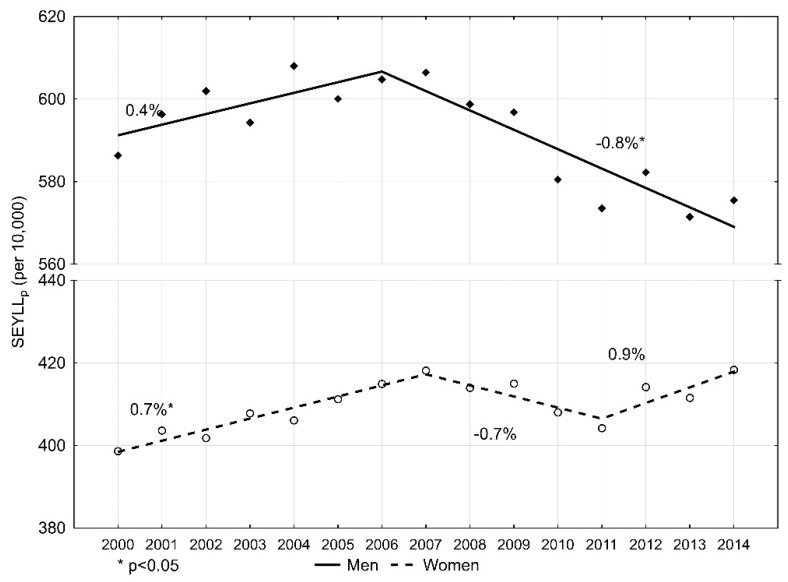
Time trends of the standard expected years of life lost per living person (SEYLL_p_) index due to malignant neoplasms in Poland, 2000–2014.

**Figure 2 ijerph-16-04898-f002:**
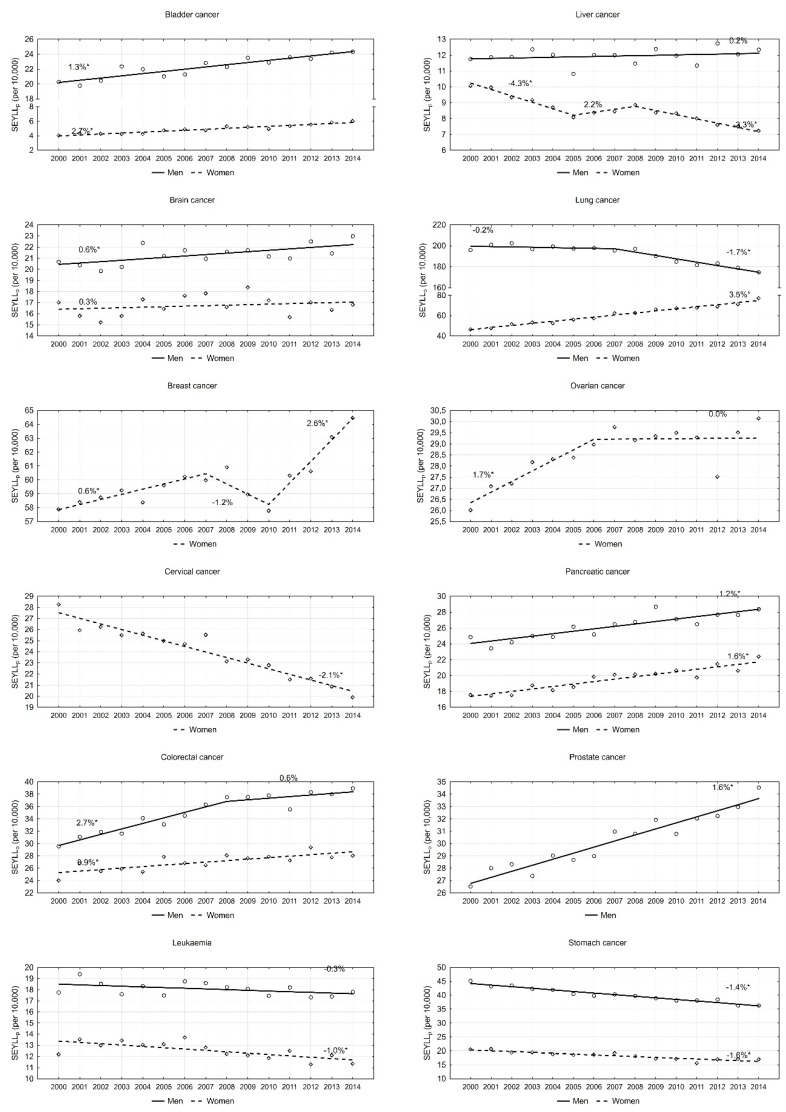
Time trends of the standard expected years of life lost per living person (SEYLL_p_) index due to the most common malignant neoplasms in Poland, 2000–2014.

**Table 1 ijerph-16-04898-t001:** Standardised death rates (per 10,000 inhabitants) due to malignant neoplasms in Poland, 2000–2014.

Causes of Death	2000	2001	2002	2003	2004	2005	2006	2007	2008	2009	2010	2011	2012	2013	2014
Men															
Malignant neoplasms overall	47.24	47.96	48.43	47.40	47.76	46.82	46.52	46.34	45.48	44.75	43.29	42.30	42.32	41.07	40.51
Bladder cancer	2.28	2.21	2.31	2.48	2.42	2.25	2.26	2.39	2.33	2.41	2.33	2.37	2.30	2.38	2.25
Brain cancer	0.97	0.99	0.78	0.83	0.92	1.09	1.07	1.07	1.11	1.14	1.06	1.04	1.12	1.05	1.03
Colorectal cancer	4.68	4.82	4.93	4.88	4.92	4.89	5.01	5.20	5.26	5.20	5.27	5.16	5.23	5.14	5.20
Leukaemia	1.13	1.23	0.99	1.03	1.06	1.31	1.31	1.32	1.29	1.24	1.21	1.31	1.24	1.26	1.23
Liver cancer	1.01	1.05	0.98	0.99	0.96	0.85	0.94	0.86	0.86	0.87	0.88	0.83	0.91	0.86	0.82
Lung cancer	15.18	15.61	12.21	11.91	12.11	14.33	14.14	13.90	13.94	13.28	12.94	12.43	12.32	11.94	11.72
Pancreatic cancer	1.78	1.76	1.80	1.82	1.79	1.79	1.80	1.84	1.87	1.90	1.82	1.73	1.82	1.77	1.83
Prostate cancer	4.16	4.46	4.57	4.32	4.46	4.36	4.38	4.50	4.31	4.34	4.08	4.17	4.20	4.15	4.11
Stomach cancer	3.95	3.77	3.78	3.60	3.53	3.34	3.23	3.23	3.11	2.98	2.95	2.86	2.78	2.66	2.63
Woman															
Malignant neoplasms overall	24.11	24.12	23.94	24.06	23.72	23.82	23.68	23.65	23.36	22.98	22.49	22.08	22.45	21.98	21.90
Bladder cancer	0.35	0.37	0.35	0.35	0.35	0.38	0.36	0.37	0.39	0.37	0.36	0.38	0.36	0.39	0.39
Brain cancer	0.71	0.65	0.59	0.62	0.66	0.79	0.85	0.83	0.80	0.86	0.81	0.75	0.80	0.76	0.69
Breast cancer	3.05	3.05	3.01	3.03	2.94	3.06	3.04	3.01	3.03	2.91	2.85	2.94	2.96	3.04	3.05
Cervix uteri cancer	1.20	1.08	1.09	1.06	1.03	1.00	1.01	1.05	0.95	0.94	0.92	0.87	0.87	0.86	0.82
Colorectal cancer	2.87	2.92	2.87	2.87	2.80	2.81	2.72	2.68	2.76	2.72	2.73	2.63	2.77	2.67	2.56
Leukaemia	0.67	0.70	0.60	0.,62	0.62	0.73	0.75	0.73	0.67	0.69	0.67	0.69	0.64	0.66	0.63
Liver cancer	0.74	0.73	0.68	0.64	0.61	0.57	0.57	0.56	0.59	0.54	0.53	0.51	0.48	0.47	0.43
Lung cancer	2.56	2.72	2.40	2.48	2.44	2.93	2.99	3.18	3.17	3.29	3.36	3.36	3.40	3.44	3.73
Ovarian cancer	1.27	1.32	1.31	1.36	1.33	1.37	1.36	1.41	1.40	1.38	1.38	1.38	1,28	1.35	1.37
Pancreatic cancer	1.26	1.23	1.23	1.29	1.20	1.21	1.28	1.29	1.33	1.25	1.25	1.22	1.29	1.24	1.29
Stomach cancer	1.48	1.47	1.37	1.35	1.28	1.20	1.19	1.20	1.18	1.06	1.05	0.97	1.00	0.99	0.97

**Table 2 ijerph-16-04898-t002:** Standard expected years of life lost in males due to malignant neoplasms in Poland, 2000–2014.

Causes of Death	SEYLL_p_ (per 10,000)		AAPC	95% CI		SEYLL_d_		AAPC	95% CI	
	2000	2014				2000	2014			
Malignant neoplasms overall	586.3	575.5	−0.3	−0.5	0.0	22.6	20.3	−0.8 *	−0.9	−0.7
Bladder cancer	20.3	24.3	1.3 *	1.0	1.7	18.8	17.0	−0.7 *	−0.8	−0.6
Brain cancer	20.7	23.0	0.6 *	0.2	1.0	31.4	27.7	−0.9 *	−1.2	−0.6
Colorectal cancer	29.5	39.0	1.8 *	1.1	2.5	20.2	18.3	−0.7 *	−1.0	−0.4
Leukaemia	17.8	17.8	−0.3	−0.7	0.0	26.9	21.4	−1.6 *	−1.9	−1.3
Liver cancer	11.8	12.3	0.2	−0.3	0.7	21.9	21.2	−0.2 *	−0.5	0
Lung cancer	196.1	174.7	−0.9 *	−1.3	−0.6	22.7	20.5	−0.7 *	−0.8	−0.6
Pancreatic cancer	24.9	28.4	1.2 *	0.8	1.6	24.0	21.4	−0.7 *	−0.8	−0.5
Prostate cancer	26.5	34.5	1.6 *	1.4	1.9	15.6	14.5	−0.5 *	−0.7	−0.3
Stomach cancer	45.2	36.3	−1.4 *	−1.6	−1.2	21.6	20.0	−0.5 *	−0.6	−0.4

SEYLL_p_, Standard Expected Years of life Lost per living persons; SEYLL_d_, Standard Expected Years of life Lost per deaths; AAPC, Average Annual Percentage Change; CI, Confidence Interval; * *p* < 0.05.

**Table 3 ijerph-16-04898-t003:** Standard expected years of life lost in females due to malignant neoplasms in Poland, 2000–2014.

Causes of Death	SEYLL_p_ (per 10,000)		AAPC	95% CI		SEYLL_d_		AAPC	95% CI	
	2000	2014				2000	2014			
Malignant neoplasms overall	398.6	418.3	0.3 *	0.1	2.0	21.5	19.4	−0.7 *	−0.8	−0.6
Bladder cancer	4.0	6.0	2.7 *	2.2	3.2	16.2	15.7	−0.2	−0.4	0.1
Brain cancer	17.0	16.8	0.3	−0.4	1.0	28.7	24.7	−1.3 *	−1.9	−0.7
Breast cancer	57.9	64.5	0.8 *	0.2	1.3	24.2	21.4	−0.9 *	−1.0	−0.8
Cervix uteri cancer	28.3	19.9	−2.1 *	−2.5	−1.8	28.0	24.3	−1.1 *	−1.5	−0.7
Colorectal cancer	24.0	28.0	0.9 *	0.5	1.3	17.9	16.4	−0.6 *	−0.8	−0.5
Leukaemia	12.2	11.3	−1.0 *	−1.5	0.4	22.9	18.5	−1.7 *	−2.0	−1.4
Liver cancer	10.1	7.2	−2.5 *	−3.4	−1.6	18.4	17.1	−0.6 *	−0.7	−0.5
Lung cancer	46.5	77.3	3.5 *	3.2	3.8	22.8	20.9	−0.5 *	−0.6	−0.4
Ovarian cancer	26.0	30.1	0.7 *	0.1	1.4	25.2	22.4	−0.9 *	−1.0	−1.1
Pancreatic cancer	17.6	22.4	1.6 *	1.2	2.0	18.7	17.6	−0.5 *	−0.6	−0.3
Stomach cancer	20.5	17.0	−1.6 *	−2.0	−1.1	18.7	18.0	−0.3 *	−0.4	−0.1

SEYLLp, Standard Expected Years of life Lost per living persons; SEYLLd, Standard Expected Years of life Lost per deaths; AAPC, Average Annual Percentage Change; CI, Confidence Interval; * *p* < 0.05.

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
