# Peer review of "Standard Expected Years of Life Lost Due to Malignant Neoplasms in Poland, 2000–2014"

_ijerph, 2019, doi:10.3390/ijerph16244898_

Round 1
Reviewer 1 Report
This is a well-prepared manuscript presenting findings from analytical research concerning a significant public health issue. The findings are relevant from the point of view of public health and cancer research.
The study was based on the epidemiological analyzes, which were described in detail in the Method section. However, I have a concern about the novelty of this study. Some data presented in Figure 2 (e.g. regarding bladder cancer) seem to be already published in another article (https://www.ncbi.nlm.nih.gov/pmc/articles/PMC5791403/ see Figure 2 and compare with Figure 2 top left the corner in the current manuscript). This requires clarification
There are some minor remarks that would require corrections to improve the quality of the paper:
1) Results (Figure 2) - because of the size, the data in figure 2 are difficult to interpret. Please consider larger text or division into 2 Figures (e.g. Figure 2 and Figure 2 cont.)
2) Discussion lines 231-233. Please re-write those sentences. This sentence should be more informative.
3) The limitations section should be provided. How about the quality of cause of death coding in Poland? In Poland, there is excessive reporting of deaths due to cardiovascular diseases. Does this affect this study? SEYLLp index is one of the ways to present the results of this study, whether using another method would affect the results? It is worth referring to this in the limitations section.
4) Some minor spell check would be helpful to improve the quality of the paper
Author Response
We are very grateful for insightful analysis of our work. We are also convinced that all the amendments will contribute to improve the quality of our manuscript.
Below are the answers for all of the Reviewer comments:
The study was based on the epidemiological analyzes, which were described in detail in the Method section. However, I have a concern about the novelty of this study. Some data presented in Figure 2 (e.g. regarding bladder cancer) seem to be already published in another article (https://www.ncbi.nlm.nih.gov/pmc/articles/PMC5791403/ see Figure 2 and compare with Figure 2 top left the corner in the current manuscript). This requires clarification.
The authors of this manuscript have been carrying out researches on epidemiology of diseases in Poland for many years. Over recent years, we have analyzed changes in mortality caused by several cancers, including bladder cancer cited by the Reviewer. Apart from two people from our team (Małgorzata Pikala and Irena Maniecka-Bryła), we have invited two urologists to cooperate. This article was published in the specialist urological journal Central European Journal of Urology.
Current manuscript is the most important our cancer research, including changes in mortality and in lost years of life due to 12 cancers being the most common causes of Poles' deaths. Therefore, the scope of our study is much broader.
There are some minor remarks that would require corrections to improve the quality of the paper:
Results (Figure 2) - because of the size, the data in figure 2 are difficult to interpret. Please consider larger text or division into 2 Figures (e.g. Figure 2 and Figure 2 cont.)
As suggested by the Reviewer, we have enlarged the font size slightly. The chart is made in high resolution, so it can be enlarged without a great loss of its quality. We are convinced that the final version of the chart will be legible.
Discussion lines 231-233. Please re-write those sentences. This sentence should be more informative.
The first sentence of the discussion was changed and the appropriate reference has been added.
The limitations section should be provided. How about the quality of cause of death coding in Poland? In Poland, there is excessive reporting of deaths due to cardiovascular diseases. Does this affect this study? SEYLLp index is one of the ways to present the results of this study, whether using another method would affect the results? It is worth referring to this in the limitations section.
The Limitations section has been added.
4) Some minor spell check would be helpful to improve the quality of the paper
Linguistic correction has been carried out.
Reviewer 2 Report
The study conducted by Pikala et al. focuses on the analysis of mortality trends due to cancer in Poland. It is a cohort study based on death certificates of more than 1 million people (1'367'364) which makes the study quite reliable and exceptional. The subject is well introduced in the introduction chapter as well as the statistical methods used for the calculation of the lost years life and those for the analysis of the time trends. However, I have some concerns regarding the discussion which is well documented but is not enough focused on the results obtained in the study. It might be helpful if the authors restructure their discussion by summarying first their results and discussed them in comparison with other similar studies performed for example in other countries. Additionally, it would benefit of some more explanation regarding specific results that have been observed in Poland and differ from those observed in other countries.
Specific comments (please see the comments in the text)
Introduction:
Lines 85-97: References are missing regarding the methods used for calculation (please see comments in the manuscript)
Discussion
Line 269: It is not clear to which study the authors refer to!
Lines 283, 296 and 308: Some explanations would be valuable in regards to the results obtained in Poland compared to the other countries.
Line 310: Why is the Polish population more threatened with cervical cancers?

Author Response
We are very grateful for insightful analysis of our work. We are also convinced that all the amendments will contribute to improve the quality of our manuscript.
Below are the answers for all of the Reviewer comments:
Introduction:
Lines 52-55: Please give more information regarding the prevention of cancers!
The part of the text in the Introduction has been added.
“The modification of risk factors plays the greatest role in primary cancer prevention. Doll and Peto estimated in their study the proportion of deaths attributed to the following risk factors in the USA: tobacco (25-40%), alcohol (2-4%), diet (10 -70%), sexual and reproductive behavior (1-13%), occupational exposure (2-8%), environmental pollution (1-5%) [10]. Another component of primary prevention is screening for early diagnosis of cervical cancer, breast cancer and colorectal cancer. Hepatitis B virus (HBV) vaccine and human papillomavirus (HPV) vaccine play an important role in cancer prevention also [11].”
Lines 85-97: References are missing regarding the methods used for calculation (please see comments in the manuscript)
The reference has been added (no 22).
Discussion
Line 269: It is not clear to which study the authors refer to!
The corresponding sentence in the Discussion has been changed.
“The study performed by the authors made the problem more evident when they revealed quite a considerable increase in the number of years of life lost due to lung neoplasms in 2000 – 2014.”
Some explanations would be valuable in regards to the results obtained in Poland compared to the other countries.
As suggested by the Reviewer, extensive parts of the Discussion section were reworded.
Lines 283: Do the author have an explanation for the increasing negative trend in Poland?
The part of the text in the Discussion has been added.
“It is associated with an increasing number of deaths due to breast cancer. In order to explain these unfavourable trends in Poland, we should point out changes in the age structure of women. Data from the National Cancer Registry indicate that most breast cancer-related deaths occur after the age of 50 (90%) [45]. The percentage of women over 50 years old in Poland in 2010 was 37.8% and it rose to 39.3% in 2014 [46].”
Lines 296: Prostate cancer
“Malignant neoplasm of the prostate is the second most common neoplasm in Poland (almost 14,000 diagnosed cases and 4,440 deaths in 2014). Wong et al. classified Poland together with Brazil, the Czech Republic, France, Ireland, Israel, Italy, Japan, Netherlands, Spain, Switzerland and the UK to the group of countries with increasing incidence and decreasing mortality due to prostate cancer [50]. However, mortality in Poland in this group was decreasing the least rapidly. The most rapid decline was observed in Israel (AAPC = -3.9), France (AAPC= -3.9%) and the Czech Republic (AAPC = -3.8%). The most positive trends are noted in three countries: Finland, Sweden and the US, where both morbidity and mortality are diminishing. Prostate cancer mostly affects older age groups. The risk of death due to this cancer increases in the seventh decade of life [51]. Males over 70 years constituted 5.9% of the population in 2000, while in 2014, the value was 7.6% of all men in Poland [46]. It can be therefore concluded that aging of the population of Poland is the main reason for the increase in prostate cancer mortality.”
Lines 308: Colorectal cancer
Poland belongs to countries with increasing morbidity and mortality rates due to colorectal cancer. Similar trends can be observed in several European countries (Bulgaria, Russia, Belarus, Estonia, Lithuania, Croatia, Spain, Latvia). The most positive trends (decreasing morbidity and mortality rates) are Europe is noted in Austria, Czech Republic, Iceland and France [52]. The rises in incidence point to the influence of dietary patterns, obesity, and lifestyle factors, whereas the mortality declines seen in more developed countries reflect improvements in survival through the adoption of best practices in cancer treatment and management in developed countries. Unfavorable incidence trends in Poland are probably resulted from the progressive adverse changes in the lifestyle and dietary patterns of Poles [53]. A continuous increase in the amount of alcohol consumed by Poles is of a concern also. In 2000, it amounted to 7.1 l per capita, while in 2014 it was 9.4 l [54]. Additionally, the reasons for a negative epidemiological situation related to colorectal neoplasms can be found in the ineffectiveness of the screening programme that has been carried out in Poland since 2000. Unfortunately, the response rate to those invitations is highly unsatisfactory. In 2014, only 17.4% of the invited respondents arrived for examinations [55].
Line 310: Why is the Polish population more threatened with cervical cancers?
The part of the discussion has been rewritten. A comparison with other European countries and two references have been added.
"Cervical cancer occurs relatively rarely in Poland. In 2012, around 2,800 females developed the disease and 1,700 females died of it. The incidence of cervical malignant neoplasms has been decreasing in Poland since the 1980s and in the last decade the decrease was even more rapid. Cervical cancer-related mortality has also been decreasing for long. It was noted by the authors of the study that this decline was also observed in the number of years of life lost due to the above disease. In the years 2000-2014 the SEYLL index was decreasing at an annual rate of -2.1%. Yet, the mortality level in Poland is still higher than in any western countries of the European Union. In addition, the rate of decline in mortality due to cervical malignant neoplasms is much more rapid in Denmark, Finland or Great Britain than in Poland, which is the reason for the increase in disparities [56]. The percentage of participants in cervical cancer screening examinations in Finland is 57%, in Holland – 70%, in Great Britain – 85%, whereas in Poland in 2015 it was only 20% [57]. Obligatory vaccination against human papilloma virus (HPV) might to some extent decrease cervical cancer-related incidence and mortality. In countries, in which vaccination against HPV is commonly applied, the number of cases of cervical cancer decreased by half [58]. Unfortunately, currently, vaccines against HPV are only recommended in Poland but they are not financed by the Ministry of Health.”
Round 2
Reviewer 1 Report
I have only one minor comment: Limitations section should be a part of the discussion (preferably at the end of this section). It should not be a separate section, especially placed after conclusions. Please transfer "limitations" to the appropriate place and start this section with a typical sentence e.g. This study has potential limitations / However, some limitations should be noted. etc.
Author Response
As suggested by the Reviewer, the Limitations section has been moved to the final part of the Discussion and has been preceded by an appropriate sentence.
Thank you to the Reviewer for a thorough analysis of our manuscript and valuable comments.
Reviewer 2 Report
The authors have satisfactorily responded to all my comments andmade the necessary changes to the manuscript. To this end, the manuscript should be accepted for publication.
Author Response
Thank you to the Reviewer for a thorough analysis of our manuscript and valuable comments.